# Transcriptomic Analysis Provides Insight into the ROS Scavenging System and Regulatory Mechanisms in *Atriplex canescens* Response to Salinity

**DOI:** 10.3390/ijms24010242

**Published:** 2022-12-23

**Authors:** Shan Feng, Beibei Wang, Chan Li, Huan Guo, Ai-Ke Bao

**Affiliations:** State Key Laboratory of Herbage Improvement and Grassland Agro-Ecosystems, Key Laboratory of Grassland Livestock Industry Innovation, Ministry of Agriculture and Rural Affairs, College of Pastoral Agriculture Science and Technology, Lanzhou University, Lanzhou 730020, China

**Keywords:** *Atriplex canescens*, salt tolerance, RNA sequencing, ROS scavenging system, protein kinases, transcription factors

## Abstract

*Atriplex canescens* is a representative halophyte with excellent tolerance to salt. Previous studies have revealed certain physiological mechanisms and detected functional genes associated with salt tolerance. However, knowledge on the ROS scavenging system and regulatory mechanisms in this species when adapting to salinity is limited. Therefore, this study further analyzed the transcriptional changes in genes related to the ROS scavenging system and important regulatory mechanisms in *A. canescens* under saline conditions using our previous RNA sequencing data. The Gene Ontology (GO) and Kyoto Encyclopedia of Genes and Genomes (KEGG) pathway annotation revealed that the differentially expressed genes (DEGs) were highly enriched in signal transduction- and reactive oxygen species-related biological processes, including “response to oxidative stress”, “oxidoreductase activity”, “protein kinase activity”, “transcription factor activity”, and “plant hormone signal transduction”. Further analyses suggested that the transcription abundance of many genes involved in SOD, the AsA-GSH cycle, the GPX pathway, PrxR/Trx, and the flavonoid biosynthesis pathway were obviously enhanced. These pathways are favorable for scavenging excessive ROS induced by salt and maintaining the integrity of the cell membrane. Meanwhile, many vital transcription factor genes (WRKY, MYB, ZF, HSF, DREB, and NAC) exhibited increased transcripts, which is conducive to dealing with saline conditions by regulating downstream salt-responsive genes. Furthermore, a larger number of genes encoding protein kinases (RLK, CDPK, MAPK, and CTR1) were significantly induced by saline conditions, which is beneficial to the reception/transduction of salt-related signals. This study describes the abundant genetic resources for enhancing the salt tolerance in salt-sensitive plants, especially in forages and crops.

## 1. Introduction

Salinity is an adverse abiotic stress that impairs the growth and development in plants, restricting the production and quality of crops and endangering the ecological environment [1,2]. Approximately one-fifth of cultivated land and one-half of irrigated land have been affected by salinization worldwide [3]. Introducing novel exogenous genes or altering the expression levels of endogenous genes is necessary to enhance the salt tolerance of plants [4]. Halophytic species growing in saline lands over a long time have developed multiple strategies to adapt to salt [1,3,5]. Accordingly, identifying the important genes and analyzing the regulatory mechanisms for salt tolerance in these species are key premises [3,6].

*Atriplex canescens*, a C_4_ semi-evergreen shrub with a strong tolerance to salt and drought stresses, is widely distributed in saline and arid regions [7]. It is used to reclaim marginal lands and is also an excellent forage for many livestock [8,9,10]. Our previous study has indicated that the addition of 100 mM NaCl (moderate salinity) could stimulate the growth of *A. canescens*, and its seedlings still maintained a certain degree of growth under 400 mM NaCl (high salinity) treatment [9]. Further physiological studies have revealed that a high capacity for photosynthesis, osmotic adjustment, and Na^+^/K^+^ homeostasis were the primary reasons for the salt tolerance of *A. canescens* [9]. According to these results, we generated transcriptome datasets of the roots and leaves of *A. canescens* to identify the important candidate genes underlying the above physiological mechanisms. We mainly focused on the differentially expressed genes (DEGs) related to ion transport, organic osmolyte accumulation, water transport, and photosynthesis [11,12]. However, salt tolerance is a complex trait involving various biochemical and physiological mechanisms, and is controlled by multiple genes. In addition to the abovementioned functional genes, the genes involved in mechanisms, such as the antioxidant systems and signaling regulatory networks, play essential roles in the response of plants to soil salinity [13,14].

The level of reactive oxygen species (ROS) in plants is low under normal growth conditions, but it increases significantly under stress conditions [15]. Numerous studies have shown that salt stress can cause ionic stress and osmotic stress in plant cells, which further results in the production of ROS [16]. High levels of ROS in plant cells trigger programmed cell death [17]. In order to maintain the balance between ROS production and scavenging, plants have developed a complex ROS scavenging system including enzymatic and non-enzymatic (antioxidants) systems [18]. Non-enzymatic systems, such as glutathione, carotenoids and phenolics, react directly with ROS by scavenging them. Enzymatic systems include superoxide dismutase, peroxidase, and catalase, which can eliminate superoxide and hydrogen peroxide [18]. It has been reported that flavonoids are the major non-enzymatic antioxidant produced in the stressed plant, and they have the ability to scavenge ROS molecules such as superoxide, hydroxyl radicals, and hydrogen peroxide (H_2_O_2_) [19]. Although studies have indicated that the high ROS scavenging capacity is also a considerable physiological trait of strong salt tolerance in *A. canescens* [20], the potential molecular basis has not yet been explored.

On the other hand, previous studies have reported that some genes are responsible for the reception/transduction of salt-related signals in plants and the regulation of downstream stress-responsive genes [4,21]. They include genes encoding various transcription factors (TFs), protein kinases, and ubiquitin-mediated proteolysis machinery [22]. However, previous studies on the molecular basis of salt tolerance in plants have mainly focused on the salt tolerance mechanisms of the model plant *Arabidopsis* and conventional crops [23]. Halophytes have evolved many unique strategies for adaptation to saline environments and may contain superior stress-resistant genes [2]. Nevertheless, knowledge on the regulatory mechanisms for *A. canescens’* salt tolerance is also limited. Thus, the screening and identifying of genes related to its adversity response regulatory network is of great significance for comprehensively elucidating the mechanism of this species in adapting to a salty environment and improving the salt tolerance of crops.

Therefore, to obtain a more complete picture of the molecular mechanisms for adapting to salinity in *A. canescens*, we further analyzed the DEGs related to ROS scavenging, transcription factors, and protein kinases by using the previous transcriptome data from *A. canescens* leaves and roots under 100 mM NaCl treatment. Finally, candidate genes were selected and qRT-PCR was used to verify the expression pattern.

## 2. Results

### 2.1. Transcriptome Profile of Leaves and Roots under NaCl Treatments

To learn more about how *A. canescens* adapts to a saline environment, we further analyzed the transcriptome data of the leaves and roots of seedlings treated with 100 mM NaCl treatments for 6 and 24 h [12]. The method of determining the DEGs was as described above. A total of 36,800 DEGs (CL6-vs-SL6, CL24-vs-SL24, CR6-vs-SR6, and CR24-vs-SR24) were identified in four comparisons: ‘C’ and ‘S’ denote the control group and salt treatment group, respectively; ‘L’ and ‘R’ denote the leaves and roots of seedlings, respectively, and ‘6’ and ‘24’ denote treatment for 6 and 24 h, respectively [12]. According to the Venn diagram, 4506 and 6128 DEGs were specifically expressed in the leaves and roots at 6 h, while 10,178 DEGs at 6 h showed overlapping expression in both the leaves and roots, indicating that signal pathways controlling these responses in *A. canescens* were interacting in different tissues (Figure 1A). Similarly, there were 2728 and 1730 DEGs specifically expressed in the leaves and roots at 24 h, while 675 DEGs showed overlapping expression in both the leaves and roots (Figure 1B). Overall, the results demonstrated that the number of DEGs at 6 h after NaCl treatment was significantly higher than at 24 h, indicating that the regulatory genes may be more complicated at an earlier time point.

### 2.2. Functional Analysis of DEGs

To study the biological pathways and molecular functions of DEGs in *A. canescens* under salt treatments, we performed a functional enrichment analysis for the GO and KEGG pathways in the DEGs at each time point for CL vs. SL and CR vs. SR. The GO annotation indicated that the DEGs at different stages and in different organs were enriched in many biological processes, cellular components, and molecular functions. At 6 h in the leaves and roots, the genes related to ROS functions were enriched among the DEGs (e.g., “response to oxidative stress”, “oxidoreductase activity”, “response to oxygen-containing compound”, and “oxidation-reduction process”) (Figure 2A,B). In particular, the biological processes related to “carotenoid metabolic process”, “carotenoid biosynthetic process”, “response to reactive oxygen species”, and “response to stimulus” were significantly enriched in DEGs at 6 h (Figure 2A). In addition, the molecular functions “oxidoreductase activity, acting on the CH-OH group of donors, oxygen as acceptor”, “peroxidase activity”, and “antioxidant activity” were enriched at 6 h in both the leaves and roots (Figure 2B, Appendix A). At 24 h, the terms “response to stimulus”, “cell communication”, “signal transduction”, “regulation of biological process”, and “biological regulation” were only significantly enriched in the leaves (Figure 3A), while “metabolic process”, “cellular metabolic process”, “biosynthetic process”, and “protein metabolic process” were obviously enriched in the roots (Figure 3A). In addition, the molecular functions “kinase activity”, “protein kinase activity”, “calcium ion binding”, and “peroxidase activity” were enriched in the leaves, and “signaling receptor activity”, “lyase activity”, and “acetyltransferase activity” were significantly enriched in the roots (Figure 3B, Appendix A). The above results indicated that ROS metabolism and regulatory genes may play crucial roles in *A. canescens*’ response to salt stress.

For the KEGG pathway, 129, 129, 123, and 123 pathways were identified from the pairwise comparisons between CL6-vs-SL6, CR6-vs-SR6, CL24-vs-SL24, and CR24-vs-SR24, respectively. Based on the ranking of the top 30 pathways, “phenylpropanoid biosynthesis”, “plant hormone signal transduction”, “carotenoid biosynthesis”, and “glutathione metabolism” were significantly enriched in CL6-vs-SL6 and CR6-vs-SR6 (Figure 2C,D, Appendix A). In addition, “glycosaminoglycan degradation”, “peroxisome”, “glycerophospholipid metabolism”, and “arachidonic acid metabolism” were significantly enriched in CL24-vs-SL24 and CR24-vs-SR24 (Figure 3C,D, Appendix A). These results demonstrated that salt treatment significantly induced the expression of genes involved in stress response production.

### 2.3. DEGs Involved in the ROS Scavenging System

The ascorbate-glutathione (AsA-GSH) cycle, the glutathione peroxidase (GPX) pathway, the catalase (CAT) pathway, the peroxiredoxin/thioredoxin (PrxR/Trx) pathway, and superoxide dismutase (SOD) together constitute the ROS scavenging system of plants [24]. This powerful ROS scavenging system is one of the vital physiological traits of strong salt tolerance in *A. canescens* [22]. In this study, the related DEGs to the ROS scavenging system were first analyzed in *A. canescens* leaves and roots after 100 mM NaCl treatment. There were 91 and 45 upregulated DEGs as well as 49 and 30 downregulated DEGs related to the ROS scavenging system in leaves when *A. canescens* seedlings were treated with 100 mM NaCl for 6 and 24 h, respectively (Figure 4A,B, Appendix A). In the four ROS scavenging pathways, GST genes were found to be involved in both the GPX pathway and the ASA-GSH pathway [25]. When plants were treated for 6 h, among the 91 upregulated DEGs, twenty-two, seventy-four, six, and four genes were classified into the AsA-GSH cycle, GPX pathway, CAT pathway, and PrxR/Trx pathway (Appendix A), respectively. Moreover, up to one-third of the upregulated DEGs showed no expression under control conditions, including six *GSTs*, two *GPXs*, fourteen *PODs*, five *GLPs*, one *CAT*, two *PEXs*, and one *SOD* (Table 1). After seedlings were treated with 100 mM NaCl for 24 h, the upregulated DEGs were still mostly categorized into the AsA-GSH pathway (27 genes) and GPX pathway (35 genes) (Appendix A), indicating that these two pathways might be important components of the ROS scavenging system in the adaptation to saline conditions in *A. canescens* leaves. Moreover, the expression of one *GLR,* one *APX*, four *GSTs*, two *GPXs*, seven *PODs*, two *GLPs*, and one *PEX* were significantly induced by salt treatment for 24 h, while they exhibited no expression in control seedlings (Table 2). Further analysis suggested that most of the upregulated DEGs were *GLP*, *POD*, and *GST* genes after salt treatment for 6 and 24 h (Figure 4A,B). More importantly, 18 DEGs were upregulated under salt treatment for both 6 and 24 h in leaves, and most of them were *POD* and *GST* genes; in particular, one *POD* (Unigene6725_All) and one *GLP* (Unigene5013_All) were almost never expressed under control conditions (Table 3).

After treatment with 100 mM NaCl for 6 h, the number of upregulated DEGs was lower than that of downregulated DEGs in the roots of *A. canescens* (Figure 4C). There were 65 upregulated DEGs, of which 14, 14, 15, and 21 genes belonged to the AsA-GSH cycle, GPX pathway, CAT pathway, and PrxR/Trx pathway, respectively, indicating that the PrxR/Trx pathway might be the primary means for *A. canescens* roots to adapt to salinity in the early period (Appendix A, Appendix A). The above upregulated DEGs included several genes encoding SOD, such as two *Fe-SODs* (CL5119.Contig3_All and Unigene5793_All) and three *Cu/Zn-SODs* (CL1360.Contig2_All, CL2699.Contig2_All, and Unigene3542_All) (Appendix A). In addition, the upregulated DEGs included one *GLR*, two *APXs*, two *GSTs*, and one *SOD*, which were rarely expressed in control plants (Table 4). After salt treatment for 24 h, the number of upregulated genes was almost equal to that of downregulated genes in the roots of *A. canescens* seedlings (Figure 4C,D). There were 29 upregulated DEGs after salt treatment for 24 h, and most of them were mainly categorized into the AsA-GSH cycle and GPX pathway, indicating that the two pathways likely play important roles in adaptation to salinity in *A. canescens* roots over a long period (Appendix A). There were two *GPX*, one *GLP,* and one *SOD* induced by salt treatment for 24 h, and the transcript abundance of these genes was not exhibited in control seedlings (Table 5). Furthermore, the greatest number of upregulated DEGs were Trxs in *A. canescens* roots under salt treatment for both 6 and 24 h (Figure 4C,D). Additionally, four DEGs were upregulated under salt treatment for both 6 and 24 h in roots, including one *GLR*, one *GST*, and two *Trxs*, indicating that these genes might take part in scavenging ROS in *A. canescens* roots under saline conditions (Table 6).

Flavonoids play a positive role in the ROS scavenging and stress signal transmission in plants [26]. In total, there were 43 upregulated DEGs and 32 downregulated DEGs categorized into the flavonoid synthesis pathway in the leaves under 100 mM NaCl for 6 h (Figure 5A). In addition, approximately a quarter of the upregulated DEGs exhibited no expression under control conditions, including one flavonol synthase (*FLS*)*,* four chalcone synthase (*CHS*), one chalcone isomerase (*CHI*), one dihydroflavonol-4-reductase (*DFR*), and seven flavonoid 3-O-glucosyltransferase (*FLG*) (Table 7). After seedlings were treated with 100 mM NaCl for 24 h, a total of twenty-two upregulated and nine downregulated DEGs were found in the leaves (Figure 5B), and among the upregulated DEGs, seven showed no expression under control conditions, but were significantly induced by salt treatment for 24 h, including two flavonoid 3-O-glucosyltransferase (*FLG*), three chalcone synthase (*CHS*), one cinnamate 4-hydroxylase (*C4H*), and one flavanone 3-hydroxylase (*F3H*) (Table 8). Moreover, under 100 mM NaCl treatment, 30 upregulated DEGs and 49 downregulated DEGs were identified in the roots of *A. canescens* at 6 h (Figure 5C), including two flavonoid 3-O-glucosyltransferase (*FLG*), three dihydroflavonol-4-reductase (*DFR*), and one flavonol synthase (*FLS*), which were upregulated in the roots of plants but almost not expressed under control conditions (Table 9). In addition, there were three cinnamate 4-hydroxylase (*C4H*) and one phenylalanine ammonia-lyase (*PAL*) with downregulated expression at 6 h after 100 mM NaCl treatment. When plants were treated for 24 h, a total of 20 upregulated DEGs were identified (Figure 5D), including nine flavonoid 3-O-glucosyltransferase (*FLG*), three dihydroflavonol-4-reductase (*DFR*), two flavanone 3-hydroxylase (*F3H*)*,* one cinnamate 4-hydroxylase (*C4H*), two flavonol synthase (*FLS*), two chalcone synthase (*CHS*), and one chalcone isomerase (*CHI*). Additionally, the transcript abundance of four flavonoid 3-O-glucosyltransferase (*FLG*), one dihydroflavonol-4-reductase (*DFR*), one chalcone isomerase (*CHI*), and one chalcone synthase (*CHS*) were not expressed in control conditions but were significantly upregulated in the roots of *A. canescens* after 100 mM NaCl treatment for 24 h (Table 10).

### 2.4. DEGs Related to Transcription Factors

The DEGs of the leaves and roots related to TFs in *A. canescens* after salt treatment were found to mainly belong to nine gene families: NAC (no apical meristem/*Arabidopsis* transcription activation factor/cup-shaped cotyledon), AP2/ERF (APETALA2 and ethylene-responsive element binding proteins), bHLH (basic helix-loop-helix), MYB (myeloblastosis), WRKY (WRKY-domain protein), ZF (zinc finger), HSF (heat shock transcription factor), bZIP (basic region-leucine zip-per/homeodomain-leucine zipper), and MAD box (Figure 6). TFs can respond rapidly to salt stress, and we mainly analyzed the DEGs of the leaves and roots related to the above TFs in *A. canescens* at 6 h.

In *A. canescens* leaves, as many as 286 TF genes were differentially expressed after exposure to 100 mM NaCl for 6 h, including 226 upregulated DEGs and 60 downregulated DEGs (Figure 6A; Appendix A). The transcripts predicted to encode WRKY1, WRKY20, WRKY56, and NAC29 were identified as upregulated, which were proved to play definite and key roles in plant salt tolerance (CL3186.Contig1_All, Unigene6298_All, Unigene1538_All, and Unigene3810, respectively) (Appendix A). The largest proportion of upregulated DEGs were ZFs, containing some C_2_H_2_-type ZFs (CL647.Contig3_All, CL7175.Contig1_All, Unigene1480_All, and Unigene251_All) (Figure 6A, Appendix A). DREBs (dehydration-responsive element binding proteins) belonged to the subfamily of AP2/ERF TFs [27]. Although few DREBs were identified in *A. canescens* leaves after salt treatment for 6 h, all of them were upregulated, including two DREB2As (CL5907.Contig1_All and Unigene3588_All) (Figure 6A; Appendix A). Moreover, the transcript abundance of five *WRKYs*, seven *MYBs*, four *bZIPs*, three *AP2/ERFs*, one *NAC*, seven *bHLHs*, three *MADS-boxes,* one *ZF,* and two *HSFs* exhibited significant upregulations which were not found in control plants; most of them were MYB proteins, particularly *MYB12* (Table 11). In addition, the transcription of six MADS-boxes was considerably induced by salt treatment, indicating their possible involvement in the process of salt adaptation in *A. canescens* leaves (Appendix A).

In *A. canescens* roots, there were 53 upregulated DEGs and 292 downregulated DEGs associated with TFs when the plants were treated with 100 mM NaCl for 6 h (Figure 6B; Appendix A). In general, the number of upregulated DEGs in the roots was much lower than those in the leaves of *A. canescens*, probably indicating that the TFs in *A. canescens* leaves play a more prominent role in its adaptation to saline conditions (Figure 6, Appendix A). The transcripts for one *MYB*, three *bZIP,* five *ZF,* one *bHLH,* one *MADS-box*, and one *HSF* genes showed no expression in control seedlings but were still significantly induced by salt, implying their important functions in *A. canescens* handling of saline conditions (Table 12). For instance, the transcript level of *HsfA2* (CL3871.Contig2_All) was upregulated more than 7-fold under 100 mM NaCl treatment, but its transcript was not found in control plants (Table 12).

### 2.5. DEGs Involved in Protein Kinases

Among protein kinases, RLKs (receptor-like protein kinases) are the main classes involved in signal perception, including LRR-RLKs (leucine-rich repeat receptor-like kinases), WAKs (cell wall-associated kinases), LecRLKs (lectin-domain-containing receptor kinases), and RLCKs (receptor-like cytoplasmic kinases) [28]. Moreover, CDPKs (calcium-dependent protein kinases) and MAPKs (mitogen-activated protein kinases) are also main classes of protein kinases, which are primarily responsible for signal transduction [25]. Therefore, the DEGs of leaves and roots related to protein kinases in *A. canescens* under 100 mM NaCl at 6 and 24 h were analyzed.

In leaves, up to 265 DEGs associated with protein kinases were identified, of which the number of upregulated DEGs was more than 2.5-fold larger than that of downregulated DEGs when *A. canescens* seedlings were treated with 100 mM NaCl for 6 h (Figure 7A, Appendix A). In particular, 46 upregulated DEGs showed no expression in the control plants, including eleven *RLKs*, five *WAKs*, seven *LecPLKs*, nineteen *LRR-RLKs*, one *MAPK*, one *LysM*, and one casein kinase (Table 13). After treatment with 100 mM NaCl for 24 h, there were 82 upregulated DEGs and 57 downregulated DEGs related to protein kinases in *A. canescens* leaves (Figure 7A, Appendix A). Among them, the transcripts levels of 23 genes were obviously upregulated under NaCl treatment but not under control conditions, and most of them were *LRR-RLKs*, *LecPLKs,* other *RLKs,* and *WAKs* (Table 14). A large proportion of the upregulated DEGs were genes predicted to encode *LRR-RLK*, *LecRLK*, *RLK*, and *WAK* proteins under salt treatment for 6 and 24 h (Figure 7A,B). Furthermore, several upregulated genes encoding different RLKs with definite roles in plant salt tolerance were found after salt treatment for either 6 or 24 h. Examples include one *FERRONIA* (Unigene13978_All), one *RLK1* (CL3313.Contig6_All), and two *CLAVATA1s* (Unigene7507_All and Unigene12732_All) (Appendix A).

In the roots, there were 59 upregulated DEGs and 49 downregulated DEGs at 6 h after salt treatment in *A. canescens* seedlings (Figure 7A, Appendix A). Some of them were not identified in control plants but showed substantial upregulation after salt treatment, such as one *RLK*, five *WAKs*, seven *LecRLKs*, eight *LRR-RLKs*, two *MAPKs*, one *Casein*, and, especially, five *RLK1s* (Table 15). Compared with salt treatment for 6 h, under treatment with salt for 24 h the number of DEGs was sharply increased, of which 73 DEGs were upregulated and 247 DEGs were downregulated (Appendix A). In particular, the transcript levels of two other *RLKs*, five *WAKs*, four *LecRLKs*, six *LRR-RLKs*, and two *CDPKs* were evidently induced by salt, while they showed no expression under control conditions (Table 16). Then, we compared DEGs in the leaves and roots of *A. canescens* under saline conditions. The results suggested that the number of genes encoding LRR-RLK was the highest under salt treatment for 6 and 24 h whether in *A. canescens* leaves or roots, and many of them were not expressed in control plants (Figure 7, Table 13, Table 14, Table 15, Table 16 and Table 17). Moreover, many transcripts of MAPK- and CDPK-encoding DEGs were induced by salt treatment in *A. canescens* leaves and roots (Figure 7). One transcript encoding CTR1 (CONSTITUTIVE TRIPLE RE-SPONSE1, CL4515.Contig1_All) was downregulated in both the leaves and roots of *A. canescens* under salt treatment for 24 h (Appendix A). In addition, several LysM and casein-encoding genes that were not previously reported to be involved in abiotic stress were also found to be upregulated in *A. canescens* leaves and roots under salt treatment (Figure 7).

### 2.6. Validation of Sequencing Data by Quantitative Real-Time PCR

To check the reproducibility of the RNA-seq results used in this study, seven DEGs were randomly selected from the leaves and roots of *A. canescens* to subsequently conduct random quantitative real-time PCR analysis. The results indicated that the fold changes in the relative expression of genes measured by quantitative reverse transcription polymerase chain reaction (qRT-PCR) were basically in agreement with the corresponding transcription abundance changes in the RNA-seq value (Appendix A). Meanwhile, the linear regression analysis of the fold changes between the RNA-seq and qRT-PCR results exhibited a high positive correlation, with R^2^ = 0.8718 in leaves and R^2^ = 0.9036 in roots under 100 mM NaCl for 6 h (Figure 8), indicating the reliability of the RNA-seq profiling data.

## 3. Discussion

Soil salinity is a global issue threatening plant growth [29]. Plants can regulate their growth in many ways to adapt to stress conditions, such as regulation of ion balance, phytohormone signaling, and the cell wall [30]. According to our previous study, a total of 36,800 DEGs were identified in four groups (CL6-vs-SL6, CR6-vs-SR6, CL24-vs-SL24, and CR24-vs-SR24), and more genes were activated efficiently in the roots and leaves at 6 h compared with 24 h of treatment to cope with salinity in *A. canescens*. Moreover, the genes related to ion transport, organic osmolyte accumulation, water transport, and photosynthesis had significant transcriptional changes in *A. canescens* under salt treatment [12]. To more comprehensively understand the potential molecular mechanisms of *A. canescens* for adapting to salinity, we analyzed the previous transcriptome data of leaves and roots in *A. canescens*. Functional analysis indicated that DEGs in the leaves and roots at 6 h were mainly involved in biological processes such as “response to oxidative stress”, “oxidoreductase activity”, “response to oxygen-containing compound”, and “oxidation-reduction process”, while DEGs mainly took part in “signal transduction”, “protein kinase activity”, and “cell communication processes” at 24 h. Given the importance of the ROS scavenging system and regulatory genes in plant responses to abiotic stress [31], we mainly focused on the ROS, TF, protein kinases, and related signaling pathways in the DEG analysis.

### 3.1. ROS Scavenging Systems Are Important for the Adaptation of A. canescens to Salinity

Saline conditions lead to ROS overproduction and subsequent oxidative stress and membrane damage in plant cells. Fortunately, plants can reduce these negative effects caused by accumulation of excessive ROS via activating efficient antioxidative defense mechanisms [32]. A previous study demonstrated that the increased activity of a variety of scavengers, including SOD, POD, and CAT, is a crucial strategy in adapting to saline conditions in *A. canescens* [20]. In the present study, many DEGs related to the ROS scavenging system were upregulated in *A. canescens* leaves and roots after 6 and 24 h of salt treatment, suggesting that *A. canescens* possesses a powerful antioxidative ability to reduce the harm triggered by ROS under 100 mM NaCl treatment. SOD acts as the first line of defense against elevated levels of ROS, taking the lead in converting superoxide anion radicals into oxygen and hydrogen peroxide (H_2_O_2_) [32]. In general, SODs can be classified into three types by their metal cofactors: copper/zinc (Cu/Zn-SOD), manganese (Mn-SOD), and iron (Fe-SOD) [33]. The activities of multiple SODs showed a significant increase when plants were exposed to saline conditions, and overexpression of several *Cu/Zn-SOD* and *Mn-SOD* genes were demonstrated to improve the salt tolerance of various plant species, such as *Nicotiana tabacum*, *Oryza sativa*, and *Arabidopsis*, hinting at their vital roles in plant abiotic stress [34,35,36]. In the present study, we found that the upregulation of several SOD-encoding genes might reduce the harmful effect of ROS in a saline environment in *A. canescens*. After the action of SOD, free oxygen radicals and hydrogen peroxide are subsequently scavenged by the AsA-GSH cycle, GPX pathway, CAT pathway, and the PrxR/Trx pathway [32]. Among all the antioxidant enzymes, the turnover rate of CAT in converting H_2_O_2_ into H_2_O and O_2_ was the highest in plant cells [37,38]. In this study, the transcript levels of three genes encoding CAT (CL9179.Contig1_All, CL9309.Contig1_All, and CL9179.Contig2_All) were increased in *A. canescens* leaves under salt treatment for either 6 or 24 h, implying that these genes might regulate the ROS homeostasis in *A. canescens’* response to salt treatment. GSH functions as a redox sensor and plays a vital role in maintaining lower levels of ROS. However, only when GSH is catalyzed by GST can it be used as an antioxidant [32,39]. Meanwhile, overexpression of the *GST/GPX* gene reduced oxidative damage in transgenic tobacco seedlings under salt treatment by increasing GSH-dependent peroxide scavenging and altering the metabolism of GSH and ASH. The above results indicated that GST might have a significant function in plant salt tolerance [40]. The present study found that many transcripts of GST in *A. canescens* were induced by salt, implying that these genes may play important roles in the adaptation to salinity of *A. canescens*, helped by GSH-dependent peroxide scavenging and metabolism of GSH and ASH. Trx is a vital member of the PrxR/Trx pathway, coupling with Prx to scavenge H_2_O_2_ [41,42]. It not only acts as an ROS scavenger but also as a vital regulator through protein–protein interactions in response to oxidative stress [24,43]. In our study, the significantly enhanced transcription level of many *Trxs* implies their essential roles in maintaining ROS homeostasis in *A. canescens* under salt treatment. PrxR is another key member of the PrxR/Trx pathway; interestingly, under 100 mM NaCl treatment, the genes encoding PrxR were merely expressed after 6 h salt treatment, but almost not at all after 24 h salt treatment, whether in the leaves or roots, indicating that PrxRs mainly contributed to reducing ROS during the short-term salt treatment of *A. canescens*. In addition, compared with the control, the enzyme activities of MDA, SOD, and POD in *A. canescens* seedlings were increased under salt stress [20].

In order to respond rapidly to stress conditions, plants have also developed non-enzymatic (antioxidant) systems in addition to the above enzymatic systems [18]. Flavonoids act as the major non-enzymatic antioxidants, becoming the secondary ROS scavenging system in plants suffering from severe stress conditions [44]. A high level of flavonoids can enhance the tolerance to abiotic stresses by inhibiting the ROS-producing enzyme and recycling several antioxidants [19]. The pathway of flavonoid biosynthesis is a deeply studied metabolic pathway, and the chalcone synthase (CHS) and chalcone isomerase (CHI) have been found to be the key enzymes of the flavonoid biosynthesis pathway. CHS was the entry point of the flavonoid pathway and catalyzed 4-Coumaroy-CoA and Malonlyl-CoA to chalcone, leading the phenylpropanoid pathway to flavonoid biosynthesis. Furthermore, CHI catalyzed chalcone to flavanone, which was further transformed into many other enzymes such as isoflavone synthase (IFS), flavanone 3-hydroxyalse (F3H), and flavonol synthase (FLS) [45]. Several key genes related to flavonoid biosynthesis, such as *CHS*, *CHI*, *F3H*, and *FLS*, were proved to enhance the accumulation of flavonoids under salt and drought stress in some plants [46]. In the present study, 11 flavonoid biosynthesis-related genes (e.g., *FLS*, *CHI*, *CHS*, and *F3H*) were identified from the transcriptome data, and most of them were markedly upregulated in the leaves and roots of *A. canescens* after salt treatment, indicating that *A. canescens* may enhance its antioxidant system by facilitating the accumulation of flavonoids. However, the role of flavonoids in the salt tolerance of *A. canescens* remains to be confirmed in the future.

### 3.2. Transcription Factors Play Significant Roles in A. canescens Response to Salt Stress by Regulating Salt Responsive Genes

TFs are vital regulators at the transcriptional level, which can directly activate or repress downstream genes by interacting with the specific *cis*-elements of their promoter region [47]. A series of TF proteins were found to play critical roles in plant salt tolerance. The WRKY family is the largest family of TFs in plants, and many of them allow plants to endure saline conditions by means of effecting multiple approaches [48]. For instance, overexpression of *Reaumuria trigyna* WRKY1 in *Arabidopsis* activated a wide range of functional genes related to the antioxidant system, proline biosynthesis, osmotic balance, and ion transport, such as *AtAPX1*, *AtCAT1*, *AtSOD1*, *AtP5CS1*, *AtP5CS2*, *AtPRODH1*, *AtPRODH2*, and *AtSOS1*, thus enhancing plant salt tolerance [49]. Similarly, the salt tolerance of transgenic plants was enhanced when *GsWRKY20* and *PsWRKY56* were overexpressed by regulating Na^+^/K^+^ homeostasis, osmoregulation, and antioxidant capacity [50,51]. In our study, the upregulation of *A. canescens WRKY1*, *WRKY20*, and *WRKY56* in leaves under salt treatment likely played indispensable roles in the salt adaptation process of *A. canescens* by mediating the regulation of stress-related genes involved in ion transport, antioxidants, and synthesis of free proline and soluble sugars. MYB proteins were also demonstrated to be key regulators in plant response to abiotic stress; *MYB12* conferred salt tolerance in *A. thaliana* by upregulating genes associated with flavonoid biosynthesis, abscisic acid (ABA) biosynthesis, proline biosynthesis, and ROS scavenging [52]. In the present study, the significant upregulation of MYB12 likely played a crucial role in adaptation to salt in *A. canescens* by controlling the expression of genes related to flavonoid biosynthesis, ABA biosynthesis, proline biosynthesis, and ROS scavenging. In addition to the above *WRKY* and *MYB* genes, other less studied *WRKYs* and *MYBs* were also obviously upregulated, such as *WRKY72*, *WRKY53*, and *MYB48*, suggesting that these genes might also play key and novel roles in *A. canescens’* salt response. *C_2_H_2_-type zinc finger ZFs* were also identified as important regulators in plant response to salt stress, because their promoter sequence contains many stress-related *cis*-acting elements [53]. In this study, the upregulation of several *C_2_H_2_-type ZFs* under saline conditions in *A. canescens* leaves suggests their possible roles in the transcriptional regulatory network in the salt-induced adaptation of *A. canescens*. Previous research has demonstrated that HSPs act as protein chaperones to alleviate the harm caused by multiple abiotic stresses in plants [54,55]. In particular, it was reported that *AtHsfA2* overexpression in *Arabidopsis* improved seedling salt tolerance [56]. In our data, the expression of a *HsfA2* was substantially upregulated under saline conditions in *A. canescens* roots, suggesting that the gene might be a key member of the regulatory network responding to salinity shock for *A. canescens*, because of its role as a protein chaperone. DREB-type TFs can be classified into two major subgroups: DREB1/C repeat binding factor (DREB1/CBF) and DREB2; of these, *DREB2* genes, especially *DREB2As*, were found to play a pivotal role in plant response to abiotic stress, including salt stress [4]. In this study, three transcripts of genes encoding DREB2A were induced by salt treatment of the leaves of *A. canescens*, implying the important roles of these genes in salt tolerance of *A. canescens*. In addition to the above TFs, NACs are also key genes in a plants’ adaptation to saline environments, especially NAC29, whose overexpression has been reported to enhance the salt tolerance of *Arabidopsis* by boosting the activity of antioxidant enzymes, including SOD and CAT [57,58]. The present study found that the transcription abundance of *A. canescens NAC29* was increased in leaves, suggesting that *AcNAC29* might influence *A. canescens*’ salt tolerance through controlling the ROS scavenging capacity. The above results suggested that *A. canescens* has evolved a strong transcriptional regulatory network when encountering a saline environment, thus enhancing its ability to adapt to salinity.

### 3.3. Protein Kinases Are Crucial in the Response to Salt of A. canescens

Currently, it is generally accepted that the transmission of stress-related signaling pathways is divided into three steps: signal perception, signal transduction, and stress-responsive gene expression [47,59]. In the above process, protein kinases control the perception and transduction of stress-related signals, many of which are key factors in plants’ response to stress [60]. Many RLKs, the largest gene family in protein kinases, were demonstrated to be important nodes in a variety of pathways of abiotic stresses [61]. For instance, recent research has indicated that overexpression of the receptor-like kinase *FERONIA (FER)*, an RLK member in the CrRLK subfamily in *Arabidopsis*, improved plants’ salt tolerance by interacting with ABI2-type and other ABI2-like phosphatases [62]. Meanwhile, under saline conditions, *Antarctic moss Pohlia nutans PnRLK 1*, a *LecRLK* gene, increased sensitivity to ABA and reduced ROS accumulation in plants, ultimately reducing the harm caused by salt stress [63]. In our data, the expression levels of several transcripts for FER and RLK1 were increased under 100 mM NaCl treatment in *A. canescens* leaves or roots, suggesting that these genes likely play an important role in the process of salt adaptation in *A. canescens* by regulating the ABA-related signal or activity of ROS scavenging. Additionally, research has shown that *Glycine max NARK*, a homolog to *A. thaliana CLAVATA1*, participates in a plant’s salt tolerance as a result of its effect on the transcription abundance of some ABA response genes [64]. In the present study, the transcripts of two *CLAVATA1s* were upregulated in the leaves under salt treatment in *A. canescens*, indicating that the two genes might confer salt tolerance on *A. canescens* through the regulation of the ABA-dependent pathway. In addition to RLK genes, other protein kinases are involved in the perception of stress signals in plants. For instance, CTR1 is a negative regulator of the gaseous hormone ethylene (ET) signaling, attributed to the involvement of proteasome-mediated degradation in key signaling components in ET. In particular, the knockout mutant *ctr1* increased the ability to tolerate salt, implying a negative correlation between CTR and plant salt tolerance [65]. In the current study, the transcription abundance of one *CTR1* was downregulated in *A. canescens* leaves and roots, indicating that the gene might also be a key factor in the salt adaptation of *A. canescens* by negatively regulating ET signaling. In the process of stress-related signal transduction, CDPK and MAPK are important participants and regulators [24]. Some *CDPK* genes alter plant salt tolerance, such as CDPK6, whose overexpression lines in Arabidopsis exhibited stronger tolerance to salt than wild-type [66,67]. MAPK pathways were also highly related to plant salt stress response. For example, a stronger capacity for coping with saline conditions emerged in *Oryza sativa MAPK5* and *MAPK44*-overexpressing lines compared with wild-type [68]. The present data illustrated that the transcript levels of many *MAPK* and *CDPK* genes were upregulated in salt-treated *A. canescens* seedlings, implying the important roles of MAPK and CDPK in signaling transduction in *A. canescens* under saline conditions. In addition, the transcripts of numerous protein kinases with uncertain roles in plant salt tolerance were also upregulated, suggesting that these genes might play novel and key roles in the mechanisms of *A. canescens*’ salt tolerance. These results indicated that *A. canescens* has developed an effective signal perception and transduction pathway when confronting saline environments.

## 4. Materials and Methods

### 4.1. Data Acquisition and Differentially Expressed Genes Analysis

Seeds of *A. canescens* were collected from plants cultivated in Lingwu County (37°78′ N, 106°25′ E; elevation 1250 m) in Ningxia Autonomous Region, China. Germination and cultivation of seedlings were carried out according to previous methods [9]. Four-week-old seedlings were divided into two groups and irrigated by the following solutions: (i) control group: 1/2 Hoagland nutrient solution; and (ii) salt treatment: 1/2 Hoagland nutrient solution with 100 mM NaCl added [12]. Data from transcriptomic analysis and eight independent gene expression libraries (CL6, SL6, CL24, SL24, CR6, SR6, CR24, and SR24) used in this study were generated as described previously [12]. In eight libraries, ‘C’ and ‘S’ denotes the control group and salt treatment group, respectively; ‘L’ and ‘R’, denote leaves and roots of seedlings, respectively, and ‘6’ and ‘24’ denote treatment for 6 and 24 h, respectively [12]. Equal amounts of total RNA were taken, pooled and were isolated from each of the four leaf tissues and each of the four root tissues and used for reverse transcription to gain a cDNA library of leaves and roots, then they were sequenced on an Illumina HiSeq™ 2000 sequencing platform in BGI Shenzhen (Shenzhen, China) [69]. After removing the low-quality tags in each library, high-quality tags were de novo assembled and clustered [70]. Eight cDNA libraries were constructed using a tag-based digital gene expression system and were further sequenced on an Illumina HiSeq™ 2000 sequencing platform (Shenzhen, China) [12,71]. All resulting clean reads were mapped to our transcriptome reference database [12]. To identify differentially expressed genes (DEGs), the number of fragments per kb per million reads (FPKM) method was used to calculate the transcript levels of all assembled unigenes [11]. Then, genes were defined as DEGs with a false discovery rate (FDR) adjusted value of *p* value < 0.001 and an absolute value of log^2^ ratio >1 as the threshold [69].

### 4.2. Regulatory Pathways Analysis of DEGs and Quantitative Real-Time PCR Validation

The BLASTX tool, with an E value ≤ 10^−5^ threshold, was used to obtain Gene Ontology (GO) annotations for unigenes, and the pathways were investigated by matching *A. canescens* genes to putative orthologs in the Kyoto Encyclopedia of Genes and Genomes (KEGG) protein database (www.genome.jp/dbget/ accessed on 10 September 2022) [11]. The GO and KEGG enrichment results were visualized using the ggplot2 package in Rstudio.

The reliability of the transcriptome analysis results in this study was verified by determining the transcript level of seven randomly selected DEGs using qRT-PCR analysis. Total RNA was extracted from control and treated samples using the TaKaRa Mini BEST Plant RNA Extraction Kit (TaKaRa, Beijing, China), and the NanoDrop ND-1000 instrument (Thermo Scientific, Waltham, MA, USA) was used to quantify the extracted RNA [12]. A total of 2 μg of DNase-treated RNA isolated from each of the four samples was further used to synthesize first-strand cDNA, in accordance with the manufacturer’s protocol (TaKaRa Biotechnology, China). qRT-PCR was performed using SYBR Green Real-Time PCR Master mix (TaKaRa Biotechnology, China) and conducted in a StepOnePlus Real-Time PCR Thermocycler (Thermo Scientific, Waltham, MA, USA). The reaction system comprised 20 μL, containing 2 μL cDNA, 10 μL SYBR Premix EX Taq II, 0.4 μL ROX Reference Dye II, 0.8 μL each of primers (10 mM), and 6 μL ddH2O. *A. canescens* ACTIN internal control gene and primer sequences used in qRT-PCR are presented in Appendix A. The relative expression levels of seven randomly selected genes were calculated using the 2^–ΔΔCt^ method [69].

## 5. Conclusions

This study presented a further and deeper analysis of the transcriptional changes in *A. canescens* in its adaptation to salt environments. The transcript levels of many DEGs associated with the ROS scavenging system were significantly upregulated, probably contributing to the effective scavenging of excessive ROS under saline conditions in *A. canescens*. Meanwhile, the upregulation of many transcription factors at the transcriptional level possibly promoted *A. canescens’* salt tolerance by regulating the expression of downstream salt stress-responsive genes. In addition, the alteration in the transcription of many candidate genes related to protein kinase might regulate the response of *A. canescens* to a saline environment by a complex network of signal transduction. This research expands our knowledge of the molecular mechanisms of salt tolerance in the halophyte species and provides a useful foundation for the genetic improvement in salt resistance of forages and crops.

## Figures and Tables

**Figure 1 ijms-24-00242-f001:**
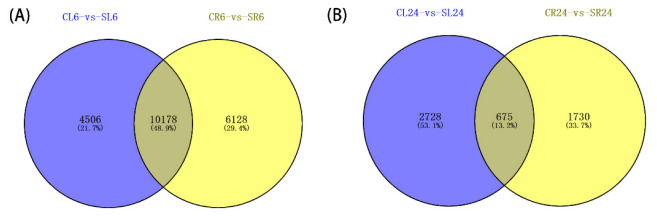
Venn diagrams showing DEGs in *A. canescens* under salt treatments. Blue and yellow colors represent leaf and root transcripts under 100 mM NaCl treatment for (**A**) 6  h and (**B**) 24  h, respectively. C and S represent the control conditions and 100 mM NaCl treatment, respectively; L and R denote the leaves and roots, respectively; and 6 and 24 denote the treatment durations.

**Figure 2 ijms-24-00242-f002:**
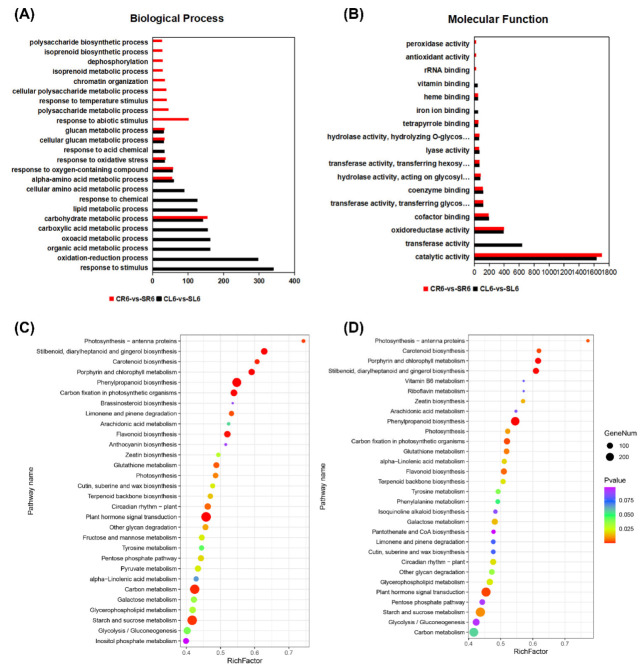
GO terms and KEGG pathways involved in the leaves and roots at 6 h in *A. canescens* under 100 mM NaCl treatment. (**A**) Biological process GO terms of CL6-vs-SL6 and CR6-vs-SR6. (**B**) Molecular function GO terms of CL6-vs-SL6 and CR6-vs-SR6. (**C**) Top 30 significantly enriched KEGG pathways in CL6-vs-SL6. (**D**) Top 30 significantly enriched KEGG pathways in CR6-vs-SR6. C and S represent the control conditions and 100 mM NaCl treatment, respectively; L and R denote the leaves and roots, respectively; and 6 denotes the treatment duration. The x-axis in (**A**,**B**) indicates the percentage of DEG numbers in each GO term; the x-axis in (**C**,**D**) indicates the percentage of DEG numbers vs. background gene numbers in each KEGG pathway.

**Figure 3 ijms-24-00242-f003:**
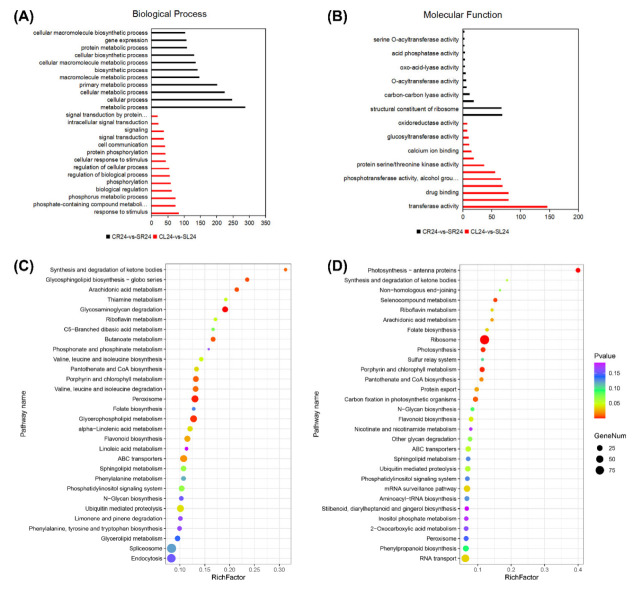
GO terms and KEGG pathways involved in the leaves and roots at 24 h in *A. canescens* under 100 mM NaCl treatment. (**A**) Biological process GO terms of CL24-vs-SL24 and CR24-vs-SR24. (**B**) Molecular function GO terms of CL24-vs-SL24 and CR24-vs-SR24. (**C**) Top 30 significantly enriched KEGG pathways in CL24-vs-SL24. (**D**) Top 30 significantly enriched KEGG pathways in CR24-vs-SR24. C and S represent the control conditions and 100 mM NaCl treatment, respectively; L and R denote the leaves and roots, respectively; and 24 denotes the treatment duration. The x-axis in (**A**,**B**) indicates the percentage of DEG numbers in each GO term; the x-axis in (**C**,**D**) indicates the percentage of DEG numbers vs. background gene numbers in each KEGG pathway.

**Figure 4 ijms-24-00242-f004:**
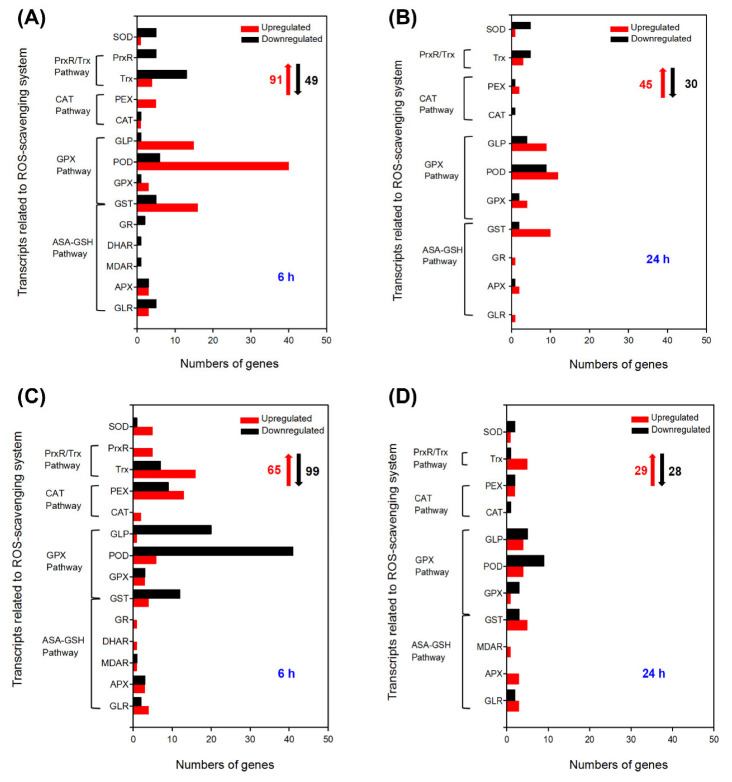
Number of DEGs related to the ROS scavenging system under 100 mM NaCl treatment for 6 and 24 h in leaves (**A**,**B**, respectively) and in roots (**C**,**D**, respectively) of *A. canescens*. SOD: superoxide dismutase, PrxR: peroxiredoxin, Trx: thioredoxin, PEX: peroxisome biogenesis, CAT: catalase, GLP: germin-like protein, GLR: glutaredoxin, POD: peroxidase, GPX: glutathione peroxidase, GST: glutathione s-transferase, GR: glutathione reductase, DHAR: dehydroascorbate reductase, MDAR: monodehydroascorbate reductase, APX: ascorbate peroxidase. The red upward arrows and black downward arrows show the total number of upregulated DEGs and downregulated DEGs, respectively.

**Figure 5 ijms-24-00242-f005:**
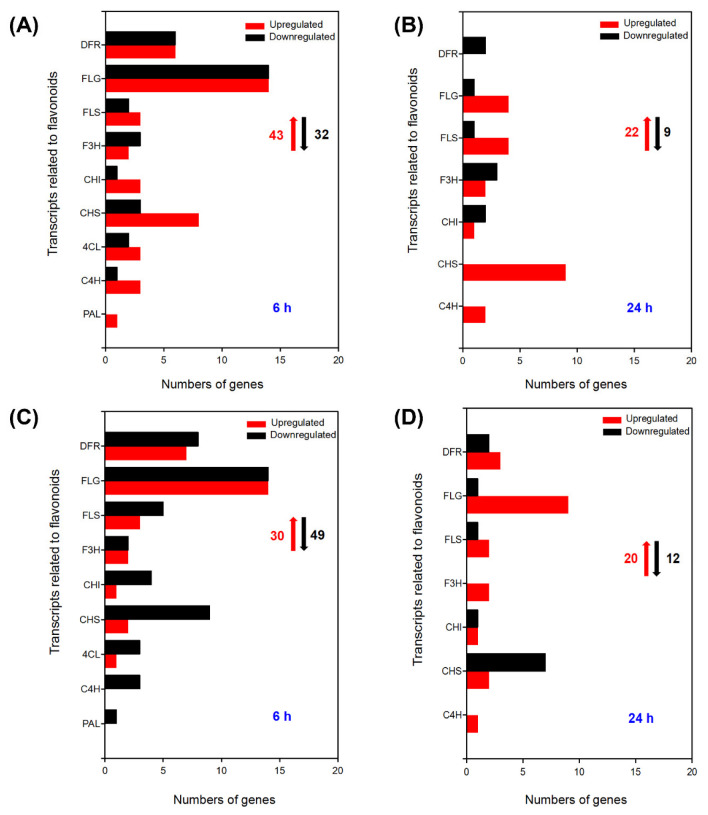
Number of DEGs related to flavonoids under 100 mM NaCl treatment for 6 h and 24 h in leaves (**A**,**B**, respectively) and in roots (**C**,**D**, respectively) of *A. canescens*. FLG: flavonoid 3-O-glucosyltransferase, DFR: dihydroflavonol-4-reductase, FLS: flavonol synthase, F3H: flavanone 3-hydroxylase, CHI: chalcone isomerase, CHS: chalcone synthase, 4CL: 4-coumarate-CoA ligase, C4H: cinnamate 4-hydroxylase, PAL: phenylalanine ammonialyase. The red upward arrows and black downward arrows indicate the total number of upregulated DEGs and downregulated DEGs, respectively.

**Figure 6 ijms-24-00242-f006:**
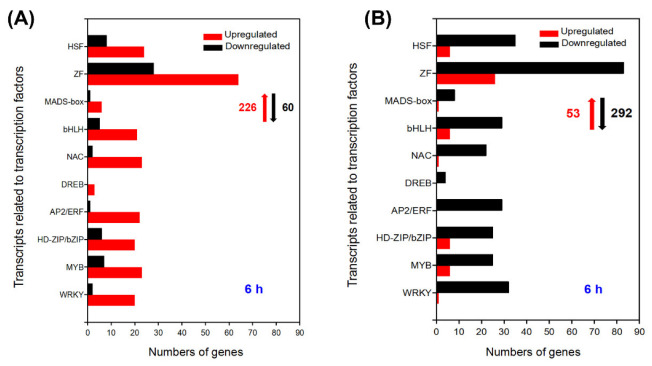
Number of DEGs related to transcription factors under 100 mM NaCl treatment for 6 h in leaves and roots (**A**,**B**, respectively) of *A. canescens*. HSF: heat shock transcription factor, ZF: zinc finger, bHLH: basic helix-loop-helix, NAC: no apical meristem/Arabidopsis transcription activation factor/cup-shaped cotyledon, DREB: dehydration responsive element binding protein, AP2/ERF: APETALA2 and ethylene-responsive element binding proteins, bZIP: basic region-leucine zip-per/homeodomain-leucine zipper, MYB: myeloblastosis, WRKY: WRKY-domain protein. The red upward arrows and black downward arrows show the total number of upregulated DEGs and downregulated DEGs, respectively.

**Figure 7 ijms-24-00242-f007:**
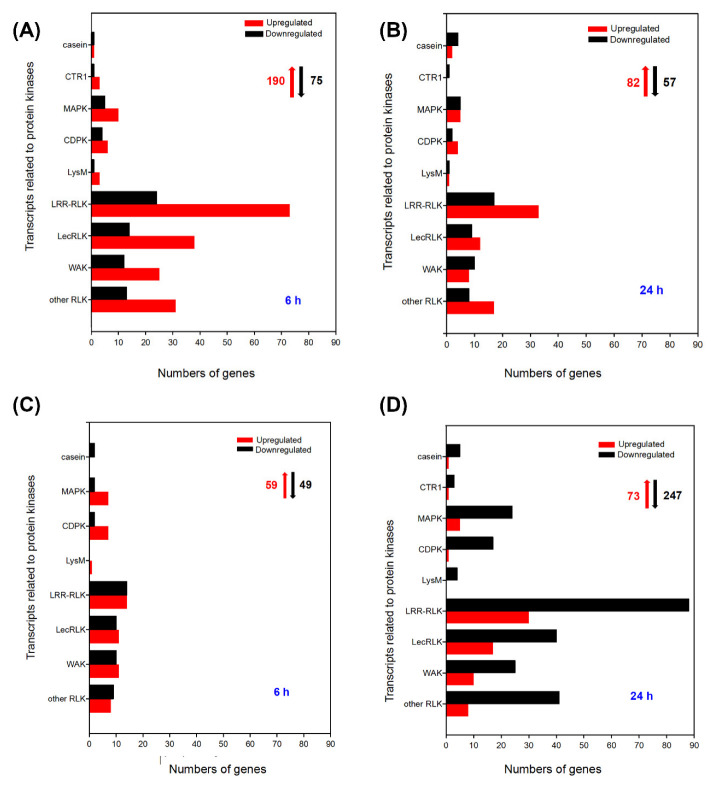
Number of DEGs related to protein kinases under 100 mM NaCl treatment for 6 and 24 h in leaves (**A**,**B**, respectively) and in roots (**C**,**D**, respectively) of *A. canescens*. CTR1: high-affinity K^+^ transporter, MAPK: mitogen-activated protein kinase, CDPK: calcium-dependent protein kinase, LysM: lysine motif, LRR-RLK: leucine-rich repeat receptor-like kinase, LecRLK: lectin-domain-containing receptor kinase, WAK: cell-wall associated kinases, RLK: receptor-like protein kinase. The red upward arrows and black downward arrows indicate the total number of upregulated DEGs and downregulated DEGs, respectively.

**Figure 8 ijms-24-00242-f008:**
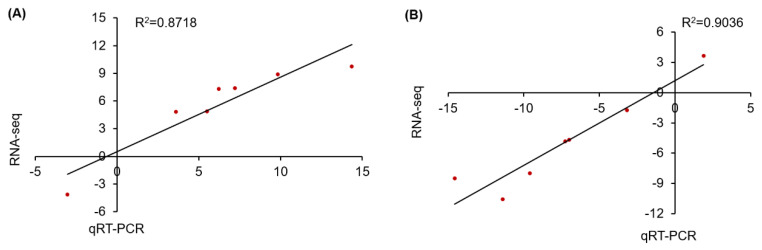
Correlation analysis for expression pattern validation of seven randomly selected DEGs under 100 mM NaCl for 6 h in leaves and in roots (**A**,**B**, respectively) by qRT-PCR. The x-axes and y-axes show the gene transcript level changes obtained by qRT-PCR and RNA-seq, respectively. R^2^ indicates the correlation.

**Table 1 ijms-24-00242-t001:** The upregulated DEGs related to the ROS scavenging system in the leaves of *A. canescens* under 100 mM NaCl but not under the control condition for 6 h.

Gene ID	Fold Change	Homologous Gene
*GST*		
Unigene39828_All	5.17	*GST-U2*
Unigene40891_All	8.79	*GST ω-2*
CL1984.Contig1_All	7.81	*GST L2*
CL2520.Contig2_All	9.82	*GSTU8*
Unigene13885_All	10.68	*GST-U2*
CL1382.Contig5_All	9.01	*GST-L*
*GPX*		
CL8077.Contig1_All	8.16	*GPX8*
CL9800.Contig2_All	8.09	*GPX*
*POD*		
Unigene8100_All	8.72	*POD57*
CL8547.Contig1_All	9.44	*POD60*
Unigene12967_All	9.58	*POD60*
Unigene4101_All	10.01	*POD57-like*
CL9189.Contig2_All	10.64	*POD2-like*
Unigene27160_All	11.00	*POD12*
Unigene13832_All	11.34	*POD39-like*
Unigene6922_All	11.80	*POD26*
Unigene3299_All	11.98	*POD57-like*
Unigene6725_All	12.86	*POD57-like*
CL1702.Contig1_All	13.37	*POD25*
CL2372.Contig2_All	10.39	*POD60*
CL4649.Contig1_All	7.81	*POD21*
CL7743.Contig2_All	10.01	*POD57-like*
*GLP*		
Unigene7967_All	9.36	*GLP*
CL3077.Contig3_All	11.58	*GLPT2*
Unigene12529_All	11.69	*GLP*
Unigene1703_All	11.88	*GLP2*
Unigene5013_All	14.63	*GLP*
*CAT*		
CL9179.Contig1_All	8.62	*CAT*
*PEX*		
CL2792.Contig2_All	7.43	*PEX1*
CL8785.Contig4_All	7.27	
*SOD*		
Unigene35175_All	7.58	*SOD*

**Table 2 ijms-24-00242-t002:** The upregulated DEGs related to the ROS scavenging system in the leaves of *A. canescens* under 100 mM NaCl but not under the control condition for 24 h.

Gene ID	Fold Change	Homologous Gene
*GLR*		
CL3957.Contig2_All	6.60	*GLR*
*APX*		
CL2142.Contig1_All	7.52	*L-APX6*
*GST*		
CL338.Contig1_All	4.09	*GST-Like*
Unigene30938_All	8.72	*GST-L2*
Unigene16620_All	3.46	*GST-Like*
CL1382.Contig1_All	3.58	*GST-Like*
*GPX*		
CL8077.Contig2_All	8.86	*GPX8*
CL4674.Contig3_All	8.73	*GPX2*
*POD*		
CL6405.Contig1_All	8.76	*POD 46-like*
CL279.Contig2_All	7.87	*POD27*
CL3665.Contig1_All	4.32	*POD*
CL5563.Contig2_All	5.73	*POD 5-Like*
Unigene31890_All	6.60	*POD48*
Unigene6725_All	4.09	*POD*
Unigene6747_All	2.81	*POD*
*GLP*		
Unigene3088_All	5.39	*GLP*
Unigene5013_All	4.46	*GLP1*
*PEX*		
CL8785.Contig7_All	5.58	*PEX10*

**Table 3 ijms-24-00242-t003:** The upregulated DEGs related to the ROS scavenging system in the leaves of *A. canescens* under 100 mM NaCl for both 6 and 24 h.

Gene ID	Homologous Gene
*APX*	
CL6261.Contig2_All	*APX6*
*GST*	
CL7491.Contig3_All	*GST U17-like*
Unigene12753_All	*GST*
Unigene30938_All	*GSTL2*
Unigene7483_All	*GST*
Unigene16620_All	*GST*
CL338.Contig1_All	*GST*
*GPX*	
CL4674.Contig3_All	*GPX2*
*POD*	
CL3665.Contig1_All	*POD*
Unigene15162_All	*POD4*
Unigene17794_All	*POD*
Unigene6747_All	*POD*
Unigene6725_All	*POD*
*GLP*	
CL2975.Contig1_All	*GLP2-1*
Unigene3088_All	*GLP*
Unigene5013_All	*GLP*
*Trx*	
Unigene1940_All	*TTL1*
Unigene5866_All	*TTL1*

**Table 4 ijms-24-00242-t004:** The upregulated DEGs related to the ROS scavenging system in the roots of *A. canescens* under 100 mM NaCl, but not under the control conditions, for 6 h.

Gene ID	Fold Change	Homologous Gene
*GLR*		
CL3957.Contig2_All	6.60	*GLR*
*APX*		
Unigene23148_All	1.02	*APX6*
CL2142.Contig1_All	0.35	*L-APX6*
*GST*		
CL7491.Contig3_All	1.40	*GST*
Unigene39828_All	0.13	*GST U2*
*SOD*		
CL5119.Contig2_All	2.62	*SOD*

**Table 5 ijms-24-00242-t005:** The upregulated DEGs related to the ROS scavenging system in the roots of *A. canescens* under 100 mM NaCl, but not under the control conditions, for 24 h.

Gene ID	Fold Change	Homologous Gene
*GPX*		
CL4674.Contig4_All	0.69	*GPX2*
CL9800.Contig1_All	6.58	*GPX*
*GLP*		
Unigene3091_All	0.65	*GLP2*
*SOD*		
CL5119.Contig2_All	5.78	*Fe-SOD*

**Table 6 ijms-24-00242-t006:** The upregulated DEGs related to the ROS scavenging system in the roots of *A. canescens* under 100 mM NaCl for both 6 and 24 h.

Gene ID	Homologous Gene
*GLR*	
Unigene2548_All	*Grx-S2*
*GST*	
CL7491.Contig3_All	*GST*
*Trx*	
Unigene6798_All	*Trx*
CL2824.Contig1_All	*Trx*

**Table 7 ijms-24-00242-t007:** The upregulated DEGs related to flavonoids in the leaves of *A. canescens* under 100 mM NaCl, but not under control conditions, for 6 h.

Gene ID	Fold Change	Homologous Gene
*CHS*		
Unigene32282_All	6.67	*CHS*
Unigene2677_All	7.80	*CHS2*
Unigene30245_All	9.85	*CHS*
CL5595.Contig2_All	11.04	*CHS*
*CHI*		
CL990.Contig2_All	6.11	*CHI*
*FLS*		
Unigene12627_All	7.37	*FLS*
*DFR*		
Unigene17164_All	7.88	*DFR-4-reductase isoform X2*
*FLG*		
CL6497.Contig2_All	7.08	*tetrahydroxychalcone glucosyltransferase*
Unigene6573_All	7.96	*flavonoid 3-O-glucosyltransferase 7*
Unigene2420_All	8.73	*hydroquinone glucosyltransferase*
Unigene32282_All	6.67	*flavonoid 3-O-glucosyltransferase 7*
Unigene6573_All	7.96	*flavonoid 3-O-glucosyltransferase 7*
Unigene29655_All	8.10	*tetrahydroxychalcone glucosyltransferase*
Unigene30587_All	8.78	*3* *′* *-O-beta-glucosyltransferase*

**Table 8 ijms-24-00242-t008:** The upregulated DEGs related to flavonoids in the leaves of *A. canescens* under 100 mM NaCl, but not under control conditions, for 24 h.

Gene ID	Fold Change	Homologous Gene
*CHS*		
Unigene33578_All	6.98	*CHS*
Unigene4107_All	5.78	*CHS*
CL3755.Contig2_All	4.17	*NAD(P)H-dependent 6* *′* *-CHS*
*F3H*		
Unigene9407_All	4.70	*F3H*
*C4H*		
CL3664.Contig1_All	7.94	*C4H*
*FLG*		
CL1972.Contig1_All	7.13	*Leucoanthocyanidin dioxygenase*
CL275.Contig3_All	5.91	*3* *′* *-O-beta-glucosyltransferase*

**Table 9 ijms-24-00242-t009:** The upregulated DEGs related to flavonoids in the roots of *A. canescens* under 100 mM NaCl, but not under control conditions, for 6 h.

Gene ID	Fold Change	Homologous Gene
*DFR*		
Unigene19482_All	5.09	*DFR4-isoform 2*
CL521.Contig8_All	10.20	*DFR4*
CL1727.Contig3_All	10.38	*DFR*
*FLS*		
Unigene19256_All	7.46	*FLS*
*FLG*		
CL6497.Contig1_All	9.66	*tetrahydroxychalcone glucosyltransferas*
CL275.Contig3_All	0.29	*3* *′* *-O-beta-glucosyltransferase*

**Table 10 ijms-24-00242-t010:** The upregulated DEGs related to flavonoids in the roots of *A. canescens* under 100 mM NaCl, but not under control conditions, for 24 h.

Gene ID	Fold Change	Homologous Gene
*DFR*		
Unigene19482_All	6.78	*DFR4-isoform 2*
*CHI*		
CL1980.Contig1_All	10.78	*CHI*
*CHS*		
Unigene32282_All	5.70	*CHS*
*FLG*		
Unigene12679_All	3.73	*flavonoid 3-O-glucosyltransferase 7*
Unigene33142_All	6.14	*anthocyanidin 5,3-O-glucosyltransferase*
CL7163.Contig1_All	6.91	*fructosephosphate glucosyltransferase*
CL275.Contig3_All	6.46	*3* *′* *-O-beta-glucosyltransferase*

**Table 11 ijms-24-00242-t011:** The upregulated DEGs related to the transcription factors in the leaves of *A. canescens* under 100 mM NaCl, but not under control conditions, for 6 h.

Gene ID	Fold Change	Homologous Gene
*WRKY*		
Unigene1538_All	7.58	*WRKY56*
Unigene40205_All	8.55	*WRKY53*
Unigene12958_All	9.76	*WRKY65*
Unigene5531_All	10.47	*WRKY61*
Unigene16243_All	11.01	*WRKY72*
*MYB*		
Unigene10098_All	7.73	*MYB 64-like*
Unigene35460_All	7.85	*PHL11-like*
Unigene28181_All	8.51	*MYB39-like*
Unigene403_All	9.71	*MYB59-like*
CL8721.Contig1_All	9.95	*MYB12*
CL1223.Contig1_All	12.14	*Myb48*
CL4043.Contig2_All	12.46	*MYB1R1*
*HD-ZIP/bZIP*		
Unigene36315_All	5.52	*bZIPATHB-51 isoform X2*
Unigene27471_All	6.93	*bZIP61*
Unigene16296_All	9.08	*bZIP*
Unigene14640_All	10.45	*TGA10-like*
*AP2/ERF*		
Unigene1741_All	7.56	*ERF091*
CL1411.Contig1_All	10.59	*BBM2*
Unigene3750_All	9.80	*PLT2*
*NAC*		
Unigene167_All	10.44	*NAC18*
*bHLH*		
Unigene1766_All	11.52	*bHLH118*
Unigene16064_All	11.30	*bHLH25*
Unigene7013_All	8.23	*bHLH086*
CL9130.Contig1_All	7.56	*bHLH25*
Unigene27074_All	6.75	*bHLH121*
Unigene10520_All	10.72	*bHLH20*
Unigene4053_All	8.05	*UPBEAT1-like*
*MAD-box*		
Unigene24561_All	9.98	*MADS18*
Unigene9423_All	9.54	*MADS23*
CL5411.Contig5_All	10.95	*MADS23*
*ZF*		
CL7398.Contig3_All	11.98	*ZF CCHC 10-like*
*HSF*		
CL3372.Contig2_All	11.37	*HSF24*
CL7682.Contig2_All	9.95	*HSF B-3-like*

**Table 12 ijms-24-00242-t012:** The upregulated DEGs related to transcription factors in the roots of *A. canescens* under 100 mM NaCl, but not under control conditions, for 6 h.

Gene ID	Fold Change	Homologous Gene
*MYB*		
Unigene13756_All	8.22	*MYB98*
*bZIP*		
Unigene23519_All	6.64	*bZIP HDG5-like*
Unigene23520_All	6.70	*bZIP HDG5-like*
Unigene26257_All	8.36	*bZIP HDG5-like*
*bHLH*		
CL7924.Contig1_All	8.77	*bHLH20*
*MAD-box*		
CL1478.Contig3_All	6.02	*AGL24-like*
*ZF*		
Unigene5674_All	7.53	*ZF domain-containing protein 7*
CL2069.Contig1_All	10.85	*ZF*
CL4799.Contig2_All	7.33	*ZF-GIS2*
CL37.Contig3_All	7.58	*ZF*
CL6690.Contig2_All	12.96	*ZF16*
*HSF*		
CL3871.Contig2_All	7.40	*HSF factor A2 isoform 4*

**Table 13 ijms-24-00242-t013:** The upregulated DEGs related to protein kinases in the leaves of *A. canescens* under 100 mM NaCl, but not under control conditions, for 6 h.

Gene ID	Fold Change	Homologous Gene
*Other RLK*		
CL2170.Contig3_All	4.39	*receptor-like protein kinase At3g47110*
CL2893.Contig5_All	6.00	*receptor-like protein kinase*
Unigene38755_All	6.08	*proline-rich receptor-like protein kinase PERK15*
CL7451.Contig5_All	6.13	*receptor-like protein kinase*
Unigene33490_All	6.70	*receptor-like protein kinase 2*
CL6175.Contig1_All	7.06	*serine/threonine-protein kinase PBL19*
Unigene34019_All	7.79	*receptor-like protein kinase*
Unigene36094_All	8.32	*receptor-like protein kinase*
Unigene21989_All	10.58	*receptor-like protein kinase HSL1*
Unigene1080_All	4.80	*receptor protein kinase TMK1*
Unigene30353_All	7.10	*receptor protein kinase*
*WAK*		
Unigene36859_All	6.16	*wall-associated receptor kinase 3-like*
Unigene29472_All	6.95	*wall-associated receptor kinase-like 16*
Unigene34082_All	7.48	*wall-associated receptor kinase-like 9*
CL8521.Contig2_All	7.95	*wall-associated receptor kinase-like 1*
Unigene35516_All	8.47	*wall-associated receptor kinase-like 16*
*LecRLK*		
Unigene21219_All	5.86	*L-type lectin-domain containing receptor kinase S.6*
Unigene26968_All	6.69	*G-type lectin S-receptor-like serine/threonine-protein kinase SD1-29 isoform X2*
Unigene37389_All	7.16	*G-type lectin S-receptor-like serine/threonine-protein kinase SD2-5*
Unigene34574_All	8.07	*L-type lectin-domain containing receptor kinase S.6-like*
CL3763.Contig1_All	8.31	*L-type lectin-domain containing receptor kinase IX.1*
CL7452.Contig2_All	9.60	*G-type lectin S-receptor-like serine/threonine-protein kinase At1g11330-like*
Unigene18214_All	12.12	*L-type lectin-domain containing receptor kinase S.5*
*LRR-RLK*		
Unigene38560_All	6.32	*leucine-rich repeat receptor-like protein kinase IMK2-like*
Unigene31266_All	6.70	*leucine-rich repeat receptor protein kinase EXS precursor*
Unigene36355_All	7.11	*leucine-rich repeat receptor-like serine/threonine-protein kinase At2g24130-like*
Unigene38082_All	7.12	*leucine-rich repeat receptor-like protein kinase family protein isoform 1*
Unigene2037_All	8.74	*leucine-rich repeat receptor-like protein kinase family protein isoform 1*
Unigene12747_All	9.39	*leucine-rich repeat receptor-like protein kinase At1g35710*
CL2166.Contig3_All	11.08	*leucine-rich repeat receptor-like protein kinase At1g35710-like*
CL2166.Contig1_All	11.64	*leucine-rich repeat receptor-like protein kinase At1g35710-like*
Unigene39549_All	6.57	*LRR receptor-like serine/threonine-protein kinase At3g47570-like*
Unigene31458_All	7.04	*LRR receptor-like serine/threonine-protein kinase At4g08850*
CL1768.Contig6_All	7.26	*LRR receptor-like serine/threonine-protein kinase At3g47570-like*
Unigene30665_All	8.03	*LRR receptor-like serine/threonine-protein kinase At3g47570-like*
CL371.Contig3_All	8.15	*LRR receptor-like serine/threonine-protein kinase FLS2*
Unigene37147_All	8.77	*LRR receptor-like serine/threonine-protein kinase At3g47570*
Unigene14583_All	9.16	*LRR receptor-like serine/threonine-protein kinase RCH1*
CL6713.Contig2_All	9.74	*LRR receptor-like serine/threonine-protein kinase At1g67720*
Unigene450_All	9.93	*LRR receptor-like serine/threonine-protein kinase FLS2*
CL2166.Contig2_All	11.39	*LRR receptor-like serine/threonine-protein kinase FLS2-like*
Unigene12036_All	12.68	*LRR receptor-like serine/threonine-protein kinase GSO1-like*
*LysM*		
Unigene28415_All	8.22	*LysM domain containing receptor kinase*
*MAPK*		
Unigene30352_All	8.79	*mitogen-activated protein kinase kinase kinase A-like*
*Casein*		
CL2804.Contig4_All	8.75	*casein kinase I isoform delta-like*

**Table 14 ijms-24-00242-t014:** The upregulated DEGs related to protein kinases in the leaves of *A. canescens* under 100 mM NaCl, but not under control conditions, for 24 h.

Gene ID	Fold Change	Homologous Gene
*Other RLK*		
Unigene21989_All	4.52	*receptor-like protein kinase HSL1*
Unigene40680_All	5.49	*receptor-like protein kinase HSL1*
CL4830.Contig1_All	6.85	*receptor protein kinase ZmPK1-like*
Unigene1080_All	8.42	*receptor protein kinase TMK1*
*WAK*		
Unigene850_All	3.58	*BRASSINOSTEROID INSENSITIVE 1-associated receptor kinase 1*
Unigene36213_All	4.64	*wall-associated receptor kinase 2-like*
Unigene4688_All	8.25	*wall-associated receptor kinase-like 2-like*
*LecRLK*		
CL7452.Contig2_All	4.86	*lectin S-receptor-like serine/threonine-protein kinase At1g11330-like*
CL865.Contig6_All	5.25	*S-locus lectin protein kinase*
CL9743.Contig2_All	6.04	*S-locus lectin protein kinase family protein*
CL6802.Contig1_All	7.52	*G-type lectin S-receptor-like serine/threonine-protein kinase At1g34300*
Unigene26968_All	7.92	*G-type lectin S-receptor-like serine/threonine-protein kinase SD1-29*
Unigene3683_All	8.41	*G-type lectin S-receptor-like serine/threonine-protein kinase At4g27290*
*LRR-RLK*		
CL8471.Contig1_All	3.91	*leucine-rich repeat transmembrane protein kinase family protein isoform 1*
Unigene39444_All	5.88	*leucine-rich repeat receptor-like protein kinase At1g35710*
CL7338.Contig1_All	8.71	*CLV1-like LRR receptor kinase*
CL2166.Contig2_All	3.46	*LRR receptor-like serine/threonine-protein kinase FLS2-like*
Unigene30007_All	4.64	*LRR receptor-like serine/threonine-protein kinase At5g48740*
Unigene35311_All	5.39	*LRR receptor-like serine/threonine-protein kinase ERECTA*
Unigene31462_All	5.98	*LRR receptor-like serine/threonine-protein kinase At3g47570*
*LysM*		
Unigene40057_All	6.66	*lysM domain-containing GPI-anchored protein 2-like*
*CDPK*		
CL8340.Contig2_All	4.46	*CDPK-related kinase 4-like*
*MAPK*		
CL4776.Contig1_All	6.29	*mitogen-activated protein kinase NTF6*
CL7688.Contig1_All	8.58	*mitogen-activated protein kinase*
*Casein*		
CL2804.Contig3_All	9.28	*casein kinase I isoform delta-like*

**Table 15 ijms-24-00242-t015:** The upregulated DEGs related to protein kinases in the roots of *A. canescens* under 100 mM NaCl, but not under control conditions, for 6 h.

Gene ID	Fold Change	Homologous Gene
*Other RLK*		
Unigene39455_All	6.00	*receptor-like protein kinase At5g61350*
*WAK*		
CL342.Contig3_All	6.04	*wall-associated kinase 2*
CL219.Contig10_All	6.82	*BRASSINOSTEROID INSENSITIVE 1-associated receptor kinase 1*
Unigene22036_All	7.68	*BRASSINOSTEROID INSENSITIVE 1-associated receptor kinase 1*
Unigene4677_All	7.73	*BRASSINOSTEROID INSENSITIVE 1-associated receptor kinase 1*
Unigene2643_All	8.99	*wall-associated kinase 2*
*LecRLK*		
CL7244.Contig1_All	5.86	*G-type lectin S-receptor-like serine/threonine-protein kinase RLK1*
CL3763.Contig2_All	6.00	*L-type lectin-domain containing receptor kinase IX.1*
CL4407.Contig2_All	6.48	*G-type lectin S-receptor-like serine/threonine-protein kinase At1g11410*
CL7244.Contig2_All	7.86	*G-type lectin S-receptor-like serine/threonine-protein kinase RLK1*
CL9096.Contig2_All	8.31	*lectin S-receptor-like serine/threonine-protein kinase RLK1*
Unigene9770_All	9.45	*G-type lectin S-receptor-like serine/threonine-protein kinase RLK1*
Unigene16277_All	10.94	*L-type lectin-domain containing receptor kinase S.5-like*
*LRR-RLK*		
CL1231.Contig1_All	8.00	*leucine-rich repeat receptor-like protein kinase isoform 8*
Unigene26697_All	8.37	*leucine-rich repeat receptor-like protein kinase At4g00330-like*
CL1803.Contig4_All	12.59	*leucine-rich repeat receptor-like protein kinase At1g35710*
Unigene35311_All	6.41	*LRR receptor-like serine/threonine-protein kinase ERECTA*
Unigene32199_All	6.75	*LRR receptor-like serine/threonine-protein kinase At4g36180-like*
Unigene35817_All	7.36	*LRR receptor-like serine/threonine-protein kinase At3g47570-like*
Unigene31462_All	7.55	*LRR receptor-like serine/threonine-protein kinase At3g47570*
Unigene13207_All	8.18	*LRR receptor-like serine/threonine-protein kinase ERECTA*
*MAPK*		
CL578.Contig2_All	7.59	*Mitogen-activated protein kinase kinase kinase 5 isoform 2*
CL1892.Contig2_All	7.95	*serine/threonine-protein kinase STN8*
*casein*		
CL2804.Contig3_All	6.95	*casein kinase I isoform delta-like*

**Table 16 ijms-24-00242-t016:** The upregulated DEGs related to protein kinases in the roots of *A. canescens* under 100 mM NaCl, but not under control conditions, for 24 h.

Gene ID	Fold Change	Homologous Gene
*Other RLK*		
CL4830.Contig2_All	5.73	*receptor protein kinase ZmPK1-like*
Unigene38876_All	6.83	*receptor-like protein kinase At5g61350*
*WAK*		
CL7712.Contig2_All	5.55	*wall-associated receptor kinase 2-like*
Unigene34082_All	5.91	*wall-associated receptor kinase-like 9*
Unigene3113_All	6.23	*wall-associated receptor kinase-like 1*
Unigene36859_All	6.78	*wall-associated receptor kinase 3-like*
Unigene23885_All	6.82	*wall-associated receptor kinase-like 9-like*
*LecRLK*		
CL30.Contig6_All	5.17	*G-type lectin S-receptor-like serine/threonine-protein kinase*
Unigene2748_All	6.19	*L-type lectin-domain containing receptor kinase IV.2*
CL4407.Contig2_All	7.22	*G-type lectin S-receptor-like serine/threonine-protein kinase*
CL9743.Contig2_All	7.72	*S-locus lectin protein kinase family protein*
*LRR-RLK*		
CL5423.Contig4_All	5.78	*leucine-rich repeat receptor-like protein kinase At5g49770*
CL1231.Contig3_All	5.88	*leucine-rich repeat receptor-like protein kinase isoform 8*
Unigene39444_All	7.00	*leucine-rich repeat receptor-like protein kinase At1g35710*
CL1803.Contig2_All	4.25	*LRR receptor-like serine/threonine-protein kinase FLS2*
Unigene13207_All	5.09	*LRR receptor-like serine/threonine-protein kinase ERECTA*
Unigene2487_All	5.36	*LRR receptor-like serine/threonine-protein kinase GSO1*
*CDPK*		
CL280.Contig5_All	1.58	*Calcium-dependent protein kinase 23 isoform 3*
CL3423.Contig3_All	4.32	*Calcium dependent protein kinase 3*

**Table 17 ijms-24-00242-t017:** The upregulated DEGs related to protein kinases in the roots of *A. canescens* under 100 mM NaCl for both 6 and 24 h.

Gene ID	Homologous Gene
*Other RLK*	
Unigene9831_All	*receptor-like protein kinase HSL1*
CL3685.Contig2_All	*receptor protein kinase*
CL2893.Contig5_All	*receptor-like protein kinase*
Unigene21989_All	*receptor-like protein kinase HSL1*
Unigene1080_All	*receptor protein kinase TMK1*
*WAK*	
Unigene15340_All	*wall-associated receptor kinase-like 1*
Unigene4688_All	*wall-associated receptor kinase-like 2-like*
Unigene36213_All	*wall-associated receptor kinase 2-like*
Unigene850_All	*BRASSINOSTEROID INSENSITIVE 1-associated receptor kinase 1 precursor*
*LecRLK*	
CL2927.Contig8_All	*G-type lectin S-receptor-like serine/threonine-protein kinase At4g03230-like*
Unigene14705_All	*L-type lectin-domain containing receptor kinase IV.2*
Unigene26968_All	*G-type lectin S-receptor-like serine/threonine-protein kinase SD1-29 isoform X2*
CL7452.Contig2_All	*G-type lectin S-receptor-like serine/threonine-protein kinase At1g11330-like*
*LRR-RLK*	
Unigene30007_All	*LRR receptor-like serine/threonine-protein kinase At5g48740*
CL7967.Contig2_All	*LRR receptor-like serine/threonine-protein kinase GSO1*

## Data Availability

Data are contained within the article and Appendix A.

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
