# Peer review of "Transcriptomic Analysis Provides Insight into the ROS Scavenging System and Regulatory Mechanisms in Atriplex canescens Response to Salinity"

_ijms, 2022, doi:10.3390/ijms24010242_

Round 1

Reviewer 1 Report

Review on the manuscript “Transcriptomic Analysis Provides Insight into ROS-scavenging  System and Regulatory Mechanism in Atriplex canescens Response to Salinity”

 The authors analyzed the differentially expressed genes related to ROS-scavenging, transcription factors, and protein kinases by using the previous transcriptome data from Atriplex canescens leaves and roots under 100 mM NaCl treatment. The study is relevant and can help in elucidating the mechanism of plants to adapt to the salinity and improving the salt tolerance of crops.

However, there are serious comments:

 1)      100 mM NaCl is not a stress for this halophyte, this was shown by the authors themselves in their previous study [9]: an increase in biomass, water content and a decrease in WUE under 100 mM NaCl treatment indicate optimal conditions for Atriplex canescens growth. Stress conditions are considered when the biomass is reduced by 50% or more. This means that the authors studied the adaptation of the halophyte to salinity, but not to salt stress - this is a fundamental difference.

This does not detract from the significance of the results obtained, but the authors have to revise the Abstract, Discussion and Conclusion sections in term to adaptation to salinity, and not to salt stress (in general and for A. canescens in particular), as they did in this manuscript.

The authors also make dubious arguments for stress (lines 445-458 “…showing that PrxRs might mainly contribute to reducing ROS during short term salt stress in A. canescens. In addition, compared with control, the enzyme activities of MDA, SOD and 457 POD in A. canescens seedlings showed increasing under salt stress [20].’’, referring to his previous study [20], where it was shown that “…the malondialdehyde, SOD and POD of A. canescens seedlings were increased differently with the increase of the concentration of NaCl … treatments” (100-400 mM NaCl). It is not clear from the abstract available online whether there was an increase at 100 mM NaCl. The data on productivity and water status [9] casts doubt on this.

Conclusion, lines 622-624: “…In addition, the alteration of transcription of many candidate genes related to protein kinase might confer salt tolerance to A. canescens by facilitating the perception and transduction of salt stress related signals under salt stress.” – this is incorrect, because there is no salt stress for A. canescens

 2)      Lines 437-440 – “… Contig1_All was high expressed after 6 h of salt treatment but not under control conditions, implying that these genes might be crucial factors for A. canescens to enhance CAT activity and consequently reduce the level of H2O2 in plant tissue under salt conditions.” – the authors cannot claim this without real data on the enzyme activity

 An increase in gene expression (an increase in transcripts accumulation) is not evidence of an increase in enzyme activity. An increase in transcripts accumulation may or may not lead to an increase in protein content as a result of various post-translational modifications. Moreover, many proteins require additional activation. Protein degradation processes are also of great importance. The authors should be very careful in this aspect in discussing their results.

 Other comments:

Results. Line 99. The authors should decipher the abbreviations CL6-vs-SL6, CL24-vs-SL24, CR6-vs-SR6, CR24-vs-SR24 , since they are the first time they occur in the text (the Methods section is at the end of the manuscript)

Figure 1.  The caption for the figure is not enough. It is necessary to add a decoding of all abbreviations (CL6-vs-SL6, CL24-vs-SL24, CR6-vs-SR6, CR24-vs-SR24)

 Figure 2 - (C, D) KEGG pathways. – What is  С, and what is  D?  Also, it is necessary to add a decoding of CL6-vs-SL6, CL24-vs-SL24, CR6-vs-SR6, CR24-vs-SR24 for А and В.

 Figure 3 - same as for Fig.2

Figure 4 – it is desirable to make the same scale on the OX axis for all figures - then it will be seen in where the changes are stronger. Also, the authors should add decoding of all designations - as it was done in their previous paper [Guo, H.; Zhang, L.; Cui, Y.-N.; Wang, S.-M.; Bao, A.-K. Identification of candidate genes related to salt tolerance of the secretohalophyte Atriplex canescens by transcriptomic analysis. BMC Plant Biol. 2019, 19]

Figure 5 – same as for Fig.4

Figure 6 – same as for Fig.4

Figure 7 – same as for Fig.4

 Method. Lines 604-605. “….Total RNA was extracted from control and treated samples according to the method described previously [11].” – it is necessary to indicate by which method the RNA was extracted and give a brief description of it with a link to the previous study.

Author Response

Reviewer 1

The authors analyzed the differentially expressed genes related to ROS-scavenging, transcription factors, and protein kinases by using the previous transcriptome data from Atriplex canescens leaves and roots under 100 mM NaCl treatment. The study is relevant and can help in elucidating the mechanism of plants to adapt to the salinity and improving the salt tolerance of crops.

However, there are serious comments:

  • 100 mM NaCl is not a stress for this halophyte, this was shown by the authors themselves in their previous study [9]: an increase in biomass, water content and a decrease in WUE under 100 mM NaCl treatment indicate optimal conditions for Atriplex canescens Stress conditions are considered when the biomass is reduced by 50% or more. This means that the authors studied the adaptation of the halophyte to salinity, but not to salt stress - this is a fundamental difference.

This does not detract from the significance of the results obtained, but the authors have to revise the Abstract, Discussion and Conclusion sections in term to adaptation to salinity, and not to salt stress (in general and for A. canescens in particular), as they did in this manuscript.

Response: Thank you for your comments. According to the reviewer’s comments and suggestions, we have revised the abstract, discussion and conclusion sections in term to adaptation to salinity (line 14, 90, 92, 437, 455, 465, 490, 561, 562, 576, 602, 662-664).

The authors also make dubious arguments for stress (lines 445-458 “…showing that PrxRs might mainly contribute to reducing ROS during short term salt stress in A. canescens. In addition, compared with control, the enzyme activities of MDA, SOD and 457 POD in A. canescens seedlings showed increasing under salt stress [20].’’, referring to his previous study [20], where it was shown that “…the malondialdehyde, SOD and POD of A. canescens seedlings were increased differently with the increase of the concentration of NaCl … treatments” (100-400 mM NaCl). It is not clear from the abstract available online whether there was an increase at 100 mM NaCl. The data on productivity and water status [9] casts doubt on this.

Response: Thank you for your comments. As shown in below figure, compared with control, the POD activity and malondialdehyde content in A. canescens seedlings showed significantly increased under 100 mM NaCl treatment (Zhang et al., 2017), in addition, the SOD activity was increased differently with the increase of NaCl concentration, indicating that SOD and POD can protect the cell membrane system of A. canescens and help it to better adapt to the salinity environment.

Effects of POD, SOD activities and MDA content under NaCl and Na2SO4 treatments

Reference (Zhang et al., 2017)

Zhang, Z.-Z.; Zhan, T.; Li, Q.; Tang, D.; Li, S.-S.; Wang, C.-Y.; HE, K.-N. Physiological and biochemical responses of Atriplex canescens seedlings to salt stress. Acta Bot. Boreal-Occident. Sin. 2017, 37, 2435-2443.

Conclusion, lines 622-624: “…In addition, the alteration of transcription of many candidate genes related to protein kinase might confer salt tolerance to A. canescens by facilitating the perception and transduction of salt stress related signals under salt stress.” – this is incorrect, because there is no salt stress for A. canescens.

Response: Thank you for your comments. We have revised the incorrect part in line 662-664 on page 43.

  • Lines 437-440 – “… Contig1_All was high expressed after 6 h of salt treatment but not under control conditions, implying that these genes might be crucial factors for canescens to enhance CAT activity and consequently reduce the level of H2O2 in plant tissue under salt conditions.” – the authors cannot claim this without real data on the enzyme activity

Response: Thank you for your comments. As the reviewer’s comments and suggestions, we have deleted the description sentence about CAT (line 471-473 on page 40).

An increase in gene expression (an increase in transcripts accumulation) is not evidence of an increase in enzyme activity. An increase in transcripts accumulation may or may not lead to an increase in protein content as a result of various post-translational modifications. Moreover, many proteins require additional activation. Protein degradation processes are also of great importance. The authors should be very careful in this aspect in discussing their results.

Response: Thank you for your comments. As the reviewer’s suggested, we have revised some statements in discussion (line 464 on page 39, 471-473, 488 on page 40).

Other comments:

Results. Line 99. The authors should decipher the abbreviations CL6-vs-SL6, CL24-vs-SL24, CR6-vs-SR6, CR24-vs-SR24 , since they are the first time they occur in the text (the Methods section is at the end of the manuscript)

Response 6: Thank you for your comments. As the reviewer’s suggested, we have added the abbreviations in line 101-103 on page 3.

Figure 1.  The caption for the figure is not enough. It is necessary to add a decoding of all abbreviations (CL6-vs-SL6, CL24-vs-SL24, CR6-vs-SR6, CR24-vs-SR24)

Response 7: Thank you for your comments. As the reviewer’s suggested, we have added the decoding of all abbreviations (line 113-116 on page 3).

Figure 2 - (C, D) KEGG pathways. – What is  С, and what is  D?  Also, it is necessary to add a decoding of CL6-vs-SL6, CL24-vs-SL24, CR6-vs-SR6, CR24-vs-SR24 for А and В.

Response 8: Thank you for your comments. According to the reviewer’s comments and suggestions, we have revised the caption for figure 2 (C) and (D), and added the decoding of CL6-vs-SL6 and CR6-vs-SR6 (line 154-157 on page 5).

Figure 3 - same as for Fig.2

Response 9: Thank you for your comments. We have revised the caption for figure 3 (C) and (D), and added the decoding of CL24-vs-SL24 and CR24-vs-SR24 (line 162-165 on page 6).

Figure 4 – it is desirable to make the same scale on the OX axis for all figures - then it will be seen in where the changes are stronger. Also, the authors should add decoding of all designations - as it was done in their previous paper [Guo, H.; Zhang, L.; Cui, Y.-N.; Wang, S.-M.; Bao, A.-K. Identification of candidate genes related to salt tolerance of the secretohalophyte Atriplex canescens by transcriptomic analysis. BMC Plant Biol. 2019, 19]

Response 10: Thank you for your comments. As the reviewer’s suggested, we revised all the figures to make them have the same scale on the OX axis and added decoding of all designations (line 198-202 on page 8).

Figure 5 – same as for Fig.4

Response 11: Thank you for your comments. As the reviewer’s suggested, we have revised all the figures same as for Fig.4 (line 273-275 on page 16).

Figure 6 – same as for Fig.4

Response 12: Thank you for your comments. As the reviewer’s suggested, we have revised all the figures same as for Fig.4 (line 331-334 on page 20).

Figure 7 – same as for Fig.4

Response 13: Thank you for your comments. As the reviewer’s suggested, we have revised all the figures same as for Fig.4 (line 391-394 on page 25).

Method. Lines 604-605. “….Total RNA was extracted from control and treated samples according to the method described previously [11].” – it is necessary to indicate by which method the RNA was extracted and give a brief description of it with a link to the previous study.

Response 14: Thank you for your comments. According to the reviewer’s comments and suggestions, the method of RNA extraction has been added in line 641-644 on page 43.

Reviewer 2 Report

The manuscript lacks the novelty as the work in the previous paper “Guo, H.; Zhang, L.; Cui, Y.-N.; Wang, S.-M.; Bao, A.-K. Identification of candidate genes related to salt tolerance of the secretohalophyte Atriplex canescens by transcriptomic analysis. BMC Plant Biol. 2019, 19” seems to overlap with the current data. The extent of prior work informing this paper cannot fully support the idea. The manuscript is not suitable for consideration for publication at this stage.

Author Response

Reviewer 2

The manuscript lacks the novelty as the work in the previous paper “Guo, H.; Zhang, L.; Cui, Y.-N.; Wang, S.-M.; Bao, A.-K. Identification of candidate genes related to salt tolerance of the secretohalophyte Atriplex canescens by transcriptomic analysis. BMC Plant Biol. 2019, 19” seems to overlap with the current data. The extent of prior work informing this paper cannot fully support the idea. The manuscript is not suitable for consideration for publication at this stage.

Response: Thank you for the comments. The main focuses are actually different between the two articles. Our previous paper mainly focuses on the genes related to ion transport, organic osmolyte accumulation, water transport and photosynthesis. In current work, we further explored the genes related to ROS-scavenging, transcription factors, and protein kinases. Atriplex canescens is a representative halophyte with excellent tolerance to salt, so screening and identifying genes related to the adversity response regulatory network from A. canescens is of great significance for comprehensively understanding the mechanism of this species to adapt to the salt environment. Thus, we want to interpret other potential molecular basis of salt tolerance in A. canescens more comprehensively through current work.

Round 2

Reviewer 2 Report

The manuscript explores the ROS scavenging system and other regulatory mechanisms in Atriplex canescens response to salinity only based on transcriptome data so it is needed to validate through experimental analysis (like antioxidant enzyme activities) to prove the hypothesis. 

Author Response

Reviewers 2

The manuscript explores the ROS scavenging system and other regulatory mechanisms in Atriplex canescens response to salinity only based on transcriptome data so it is needed to validate through experimental analysis (like antioxidant enzyme activities) to prove the hypothesis.

Response: Thank you for your comments. According to previous literature, we found that similar work on the accumulation of the different types of ROS and the activities of ROS scavenging system in A. canescens response to salinity have been done by another group (Zhang et al., 2017). As shown in below figure, the malondialdehyde, SOD and POD of A. canescens seedlings were increased differently with the increase of the concentration of NaCl treatments and the results were consistent with our transcriptome data, so we think the results of Zhang et al. and our previous research are enough to support our current conclusions. We have also discussed the results of the relevant enzyme activities in lines 477 to 478 of our manuscript.

Round 3

Reviewer 2 Report

The authors have satisfactorily addressed the concern which was raised in the revision. I would recommend the manuscript for publication.